

# Measurement of the transverse polarization of electrons emitted in neutron decay – nTRV experiment

**Kazimierz Bodek[1]⋆ and Adam Kozela[2]**

**1** M. Smoluchowski Institute of Physics, Jagiellonian University, Cracow, Poland
**2** H. Niewodniczański Institute of Nuclear Physics,
Polish Academy of Sciences, Cracow, Poland

⋆ kazimierz.bodek@uj.edu.pl

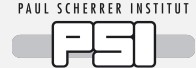

## Abstract

This paper recalls the main achievements of the nTRV experiment which measured two components of the transverse polarization ($\sigma_{T_1}$, $\sigma_{T_2}$) of electrons emitted in the $\beta$-decay of polarized, free neutrons and deduced two correlation coefficients, $R$ and $N$, that are sensitive to physics beyond the Standard Model. The value of time-reversal odd coefficient $R$, 0.004±0.012±0.005, significantly improved limits on the relative strength of imaginary scalar coupling constant in the weak interaction. The value obtained for the time-reversal even correlation coefficient $N$, 0.067±0.011±0.004, agrees with the Standard Model expectation, providing an important sensitivity test of the electron polarimeter. One of the conclusions of this pioneering experiment was that the transverse electron polarization in the neutron $\beta$-decay is worth more systematic exploring by measurements of yet experimentally not attempted correlation coefficients such as $H$, $L$, $S$, $U$ and $V$. This article presents a brief outlook on that questions.

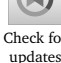
## 15.1 Introduction

Beta decay theory was firmly established about six decades ago and became a part of the Standard Model (SM). It describes the semi-leptonic and strangeness-conserving processes in the 1-st particle generation mediated by charged $W$-boson exchange. Among the empirical foundations of the electroweak sector of the SM, the assumptions of maximal parity violation, the vector and axial-vector character, and massless neutrinos are directly linked to nuclear and neutron beta decay experiments. In this way, nuclear and neutron beta decay have played a central role in the development of the weak interaction theory. Beta decay experiments with increasing precision still confirm the first two assumptions – only the neutrino masses have been shown to be finite. However, many open questions remain such as the origin of parity

violation, the hierarchy of fermion masses, the number of particle generations, the mechanism of CP violation, and the unexplained large number of parameters of the theory. The CKM matrix induced mechanism of CP violation reported for heavier systems in [1, 2] is far too weak to explain the matter-antimatter asymmetry of universe so that new CP- or T-violation sources are subject of intensive searches. In particular, interesting are processes in the systems built of light quarks with vanishingly small contributions of the CKM matrix mechanism such as nuclear beta decay. Experiments with free neutrons play a particularly important role since their interpretation is not biased with nuclear and atomic structure uncertainties. In addition, the effects of electromagnetic interaction of charged decay products in the final state (proton, electron), which can mimic T-violation, are small and can be reliably calculated [3–5].

The nTRV project at PSI, was the first experimental search for the real and imaginary parts of the scalar and tensor couplings using the measurement of the transverse polarization of electrons emitted in the free neutron decay. There are very few measurements of this observable in general [6,7], and only two in nuclear beta decays. One of them, for the $^{8}$Li system [8], provides the most stringent limit on the tensor coupling constants of the weak interaction.

According to [9], the decay rate distribution from polarized neutrons as a function of electron energy ($E$) and momentum ($\mathbf{p}$) is proportional to:

$$\omega(\mathbf{J}, \hat{\boldsymbol{\sigma}}, E, \mathbf{p}) \propto 1 + \frac{\langle \mathbf{J} \rangle}{J} \cdot \left( A\frac{\mathbf{p}}{E} + N\hat{\boldsymbol{\sigma}} + R\frac{\mathbf{p} \times \hat{\boldsymbol{\sigma}}}{E} \right) + \dots, \tag{15.1}$$

where $\frac{\langle \mathbf{J} \rangle}{J}$ ($J = |\mathbf{J}|$) is the neutron polarization, $\hat{\boldsymbol{\sigma}}$ is the unit vector onto which the electron spin is projected, and $A$ is the beta decay asymmetry parameter. $N$ and $R$ are correlation coefficients which, for neutron decay with usual SM assumptions: $C_V = C_V' = 1$, $C_A = C_A' = \lambda = -1.276$ [10] and allowing for a small admixture of scalar and tensor couplings $C_S$, $C_T$, $C_S'$, $C_T'$, can be expressed as:

$$N = -0.218 \cdot \mathrm{Re}(\mathfrak{S}) + 0.335 \cdot \mathrm{Re}(\mathfrak{T}) - \frac{m}{E} \cdot A, \tag{15.2}$$

$$R = -0.218 \cdot \mathrm{Im}(\mathfrak{S}) + 0.335 \cdot \mathrm{Im}(\mathfrak{T}) - \frac{m}{137\,p} \cdot A, \tag{15.3}$$

where $\mathfrak{S} \equiv (C_S + C_S')/C_V$, $\mathfrak{T} \equiv (C_T + C_T')/C_A$ and $m$ is the electron mass. The SM value of $N$ is finite, $N \approx -\frac{m}{E} \cdot A \approx 0.068$, and well within reach of this experiment. Its determination provides an important test of the experimental sensitivity. The $R$ correlation coefficient vanishes in the lowest order SM calculations. It becomes finite if final state interactions are included, $R_{FSI} \approx -\frac{m}{137p} \cdot A \approx 0.0006$, two orders of magnitude below the sensitivity of this experiment. A larger value of $R$ would provide evidence for the existence of exotic couplings, and a new source of time reversal violation (TRV). Using Mott polarimetry, both transverse components of the electron polarization can be measured simultaneously: $\sigma_{T_1}$ contained in the decay plane defined by the neutron spin and electron momentum associated with $N$, and $\sigma_{T_1}$ perpendicular to the decay plane and associated with $R$.

## 15.2 Experiment

The experiment was carried out at the high intensity cold neutron beam line [11] at the neutron source SINQ of the Paul Scherrer Institute, Villigen, Switzerland. The final result is based on the analysis of two data sets collected in 2006 and 2007. It profits additionally from the experience gained during another two test runs performed in 2003 and 2004. Each of those measurements featured slightly different basic conditions such as beam polarization, Mott foil thickness, spin holding magnetic field and collected statistics.

Applied detector setup was left-right symmetric. Two identical systems of detectors were arranged in planar configuration on each side of the decay volume (Figure 15.1). Each of

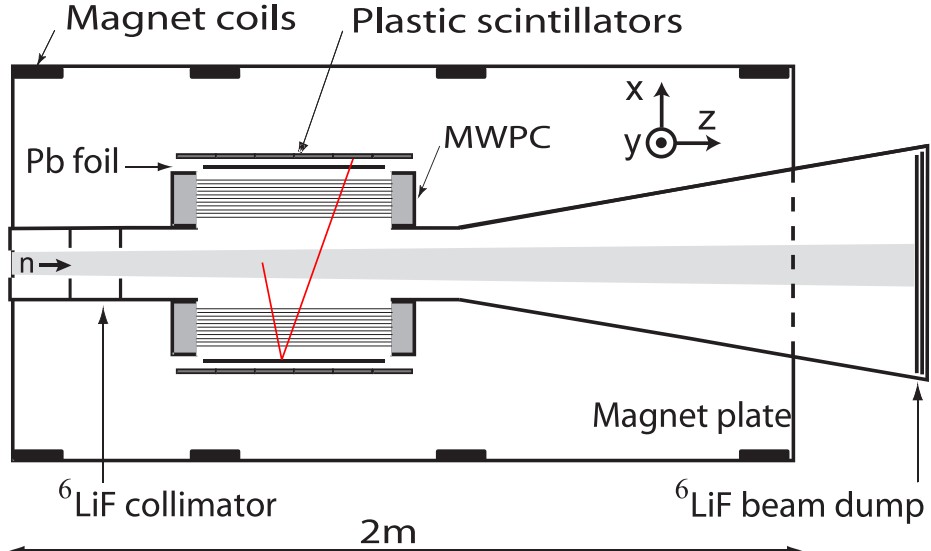

Figure 15.1: Schematic top view of the experimental setup. An electron V-track event seen by both MWPCs and scintillator on one side is indicated in red [12].

them consisted of a multi-wire proportional chamber (MWPC) for tracking of the electron trajectories and a scintillator hodoscope for electron energy measurement. Between these detectors a removable, remotely controlled, Mott scatterer (1-2 $\mu$m Pb layer evaporated on a 2.5 $\mu$m thick Mylar foil) was installed. The whole structure was mounted inside a large-volume dipole magnet providing a homogeneous vertical spin-holding field of 0.5 mT within the beam fiducial volume. A system of two RF-spin flippers (not shown in Figure 15.1) was used to control the orientation of the neutron beam polarization, reversing it at a regular time intervals, typically every 16 s.

Scintillator hodoscopes, formed by six 10-mm-thick and 60-cm-long plastic scintillator slabs were used for the electron energy reconstruction with the resolution of 33 keV at 500 keV. As the light signal produced in each segment was read out from its both ends, the up-down asymmetry has been used to determine the vertical hit position with a resolution of about 6 cm. The width of each segment (10 cm) of this hodoscope provided an approximate estimate of the position in horizontal direction (z-coordinate). Matching the position information from the MWPC and that from the scintillator hodoscope considerably helped to reduce background and random coincidences.

A 1.3-m-long multi-slit $^6$Li-based collimator defined the beam cross section to $4 \times 16$ cm$^2$ at the entrance of the Mott polarimeter. The beam was transported in pure helium at atmospheric pressure and the whole surface of the decay chamber was lined with $^6$Li loaded polymer. The total flux of the collimated beam was typically about $10^{10}$ neutrons/sec.

Dedicated measurement was performed to study the beam polarization as a function of neutron wavelength and position [11]. It showed a substantial dependence of polarization on the position in the beam fiducial volume. As a consequence the average beam polarization, necessary for the evaluation of the $N$- and $R$-correlation coefficients, was extracted from the observed decay asymmetry using the beta decay asymmetry parameter $A = -0.1196 \pm 0.0002$ [10] measured accurately in other experiments. This approach automatically accounts for the position-dependent beam density and polarization as well as for the detector acceptance. For this analysis, a large sample of single track events corresponding to electrons from neutrons decay (with only one reconstructed track segment on the triggering scintillator side) was

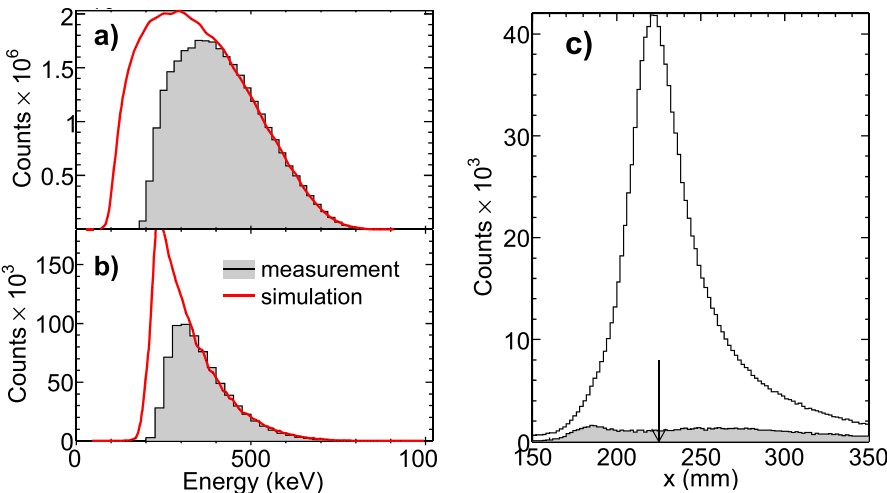

Figure 15.2: Background-corrected experimental energy distributions (shaded areas) of (a) the single-track and (b) V-track events compared with simulations. (c) Background contribution (shaded) to vertex $x$-coordinate distribution of V-track events. The arrow indicates the Mott foil position [12].

recorded, using a dedicated prescaled trigger. The main event trigger was used to identify and record all V-track candidates: events with two reconstructed segments on one side (indicating at Mott scattering from lead) and one segment accompanied by a scintillator hit on the opposite side, (see Figure 15.1). For more details on experimental setup, beam quality and performance of the detectors see [13].

The following asymmetries were analyzed to extract the beam polarization, $P$:

$$\mathcal{E}(\beta, \gamma) = \frac{N^+(\beta, \gamma) - N^-(\beta, \gamma)}{N^+(\beta, \gamma) + N^-(\beta, \gamma)} = P\beta A cos(\gamma), \tag{15.4}$$

where $N^{\pm}$ are experimental, background-corrected counts of single tracks sorted in 4 bins of the electron velocity $\beta$, and 15 bins of the electron emission angle $\gamma$ with respect to the neutron polarization direction. The sign in the superscripts reflects the beam polarization direction.

A comparison between the measured and MC simulated energy spectra for direct and Mott-scattered electrons is shown in Figure 15.2 a and b, respectively. Electronic thresholds are not included in the simulation – this is why the measured and simulated distributions do not match at the low energy side.

Another set of asymmetries was used to extract the $N$ and $R$ correlation coefficients :

$$\mathcal{A}(\alpha) = \frac{n^+(\alpha) - n^-(\alpha)}{n^+(\alpha) + n^-(\alpha)}, \tag{15.5}$$

where $n^{\pm}$ represent background-corrected experimental numbers of counts of V-track events, sorted in 12 bins of $\alpha$, the angle between electron scattering and neutron decay planes. In the case of V-track events, beside the background discussed previously, events for which the scattering took place in the surrounding of the Mott-target provide an additional source of background. Figure 15.2 c shows the distribution of the reconstructed vertex positions in the $x$-direction for data collected with and without the Mott foil. The distribution clearly peaks at the foil position. This relatively broad distribution is a result of extrapolation of two electron track segments crossing at relatively small angle ($20^o - 60^o$). Additionally, the

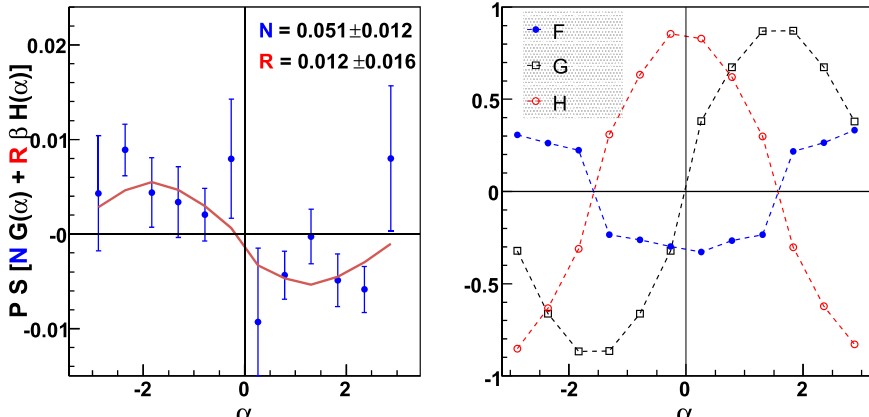

Figure 15.3: Left panel: experimental asymmetries $\mathcal{A}$ corrected for the $P\bar{\beta}A\bar{\mathcal{F}}$ term for the 2007 data set as a function of $\alpha$ (defined in text). The solid line illustrates a two-parameter $(N, R)$ least-square fit to the data. The indicated errors are statistical. Right panel: geometrical factors $\bar{\mathcal{F}}(\alpha)$, $\bar{\mathcal{G}}(\alpha)$ and $\bar{\mathcal{H}}(\alpha)$ for the same data set [12].

electron straggling effects contribute to this broadening. The "foil-out" distribution has been scaled appropriately by a factor deduced from the accumulated neutron beam.

It can be shown [13] that

$$\mathcal{A}(\alpha) - P\bar{\beta}A\bar{\mathcal{F}}(\alpha) = P\bar{S}(\alpha)\left[N\bar{\mathcal{G}}(\alpha) + R\bar{\beta}\bar{\mathcal{H}}(\alpha)\right], \tag{15.6}$$

where the kinematical factors $\bar{\mathcal{F}}(\alpha)$, $\bar{\mathcal{G}}(\alpha)$, and $\bar{\mathcal{H}}(\alpha)$ represent the average values of the quantities $\hat{\mathbf{J}}\cdot\hat{\mathbf{p}}$, $\hat{\mathbf{J}}\cdot\hat{\boldsymbol{\sigma}}$ and $\hat{\mathbf{J}}\cdot\hat{\mathbf{p}}\times\hat{\boldsymbol{\sigma}}$, respectively, $\bar{S}$ is the effective analyzing power of the electron Mott scattering, known in the literature as "Sherman function", and the bar over a letter indicates event-by-event averaging. The term $P\bar{\beta}A\bar{\mathcal{F}}$ accounts for the $\beta$-decay-asymmetry-induced nonuniform illumination of the Mott foil. Since the $\bar{\beta}$ and $\bar{\mathcal{F}}$ are known precisely from event-by-event averaging, the uncertainty of this term is dominated by the error of the average beam polarization $P$.

Mean values of the effective analyzing powers as a function of electron energy, scattering and incidence angles were calculated using the Geant 4 simulation framework [14], following guidelines presented in [15, 16]. This approach accounts properly for the atomic structure, nuclear size effects as well as the effects introduced by multiple scattering in thick foils.

The systematic uncertainty is dominated by the effects introduced by the background subtraction procedure, connected with the choice of the geometrical cuts defining event classes "from-beam" and "off-beam". To estimate this effect, the cuts were varied in a range limited solely by the geometry of the apparatus. Because the radio–frequency of the spin flippers was a small source of noise in the readout electronics, tiny spin-flipper-correlated dead time variations were observed. The result was corrected for this effect.

The asymmetries as defined in (15.4) and (15.5) have been calculated for events with energies above the neutron $\beta$-decay end-point energy and for events originating outside of the beam fiducial volume: they were found to be consistent with zero within the statistical accuracy, which proves that the data analysis was not biased e.g. with a spin-flipper-related false asymmetry.

A fit of the experimental asymmetries $\mathcal{A}$, corrected for the $P\bar{\beta}A\bar{\mathcal{F}}$ term for the experimental data set of 2007 is shown in Figure 15.3.

From the approximate symmetry of the detector with respect to the transformation $\alpha \to -\alpha$, it follows that $\bar{\beta}$, $\bar{S}$ and the factors $\bar{\mathcal{F}}$, $\bar{\mathcal{H}}$ are all symmetric, while $\bar{\mathcal{G}}$ is an antisymmetric function of $\alpha$ (see Figure 15.3). This allows the extraction of the $N$ coefficient from the expression [13]:

$$N \approx \frac{(r-1)}{(r+1)} \cdot \frac{1 - \frac{1}{2}(P\bar{\beta}A\bar{F})^2}{P\bar{S}\bar{\mathcal{G}}}, \quad r = \sqrt{\frac{n^+(\alpha)\,n^-(-\alpha)}{n^-(\alpha)\,n^+(-\alpha)}}. \tag{15.7}$$

The advantage of this method is that the impact of uncertainty of the term $P\bar{\beta}A\bar{F}$ is suppressed by a factor of about 60 compared to (15.6). The good agreement between the $N$ values obtained in both ways enhances confidence in the extracted $N$ and $R$ coefficient values.

The systematic uncertainties in the evaluation of the $R$ and $N$ coefficients are dominated by effects introduced by the background subtraction procedure and the choice of specific values of the cuts that determine whether an individual event is attributed to "signal" or to "background". The impact of these effects was systematically studied for all data sets. Another systematics is related to the Mott-target mass distribution as it can influence the electron depolarization leading to increased uncertainty of the effective Sherman function. Additional calibration measurements were performed to determine the Mott-target thickness distribution using the photon induced X-ray emission method [17]. A detailed description of the data analysis process can be found in [12, 18] together with the final result comprising all available experimental data.

$$N = 0.067 \pm 0.022_{\text{stat}} \pm 0.004_{\text{syst}}, \tag{15.8}$$

$$R = 0.004 \pm 0.012_{\text{stat}} \pm 0.005_{\text{syst}}. \tag{15.9}$$

This was the first determination of the $N$ correlation coefficient in $\beta$-decay.

In Figure 15.4 the new results are included in exclusion plots containing all experimental information available from nuclear and neutron beta decays as surveyed in [19]. The upper plots contain the normalized scalar and tensor coupling constants $\mathfrak{S}$ and $\mathfrak{T}$, while the lower plots correspond to the helicity projection amplitudes in the leptoquark exchange model, as defined in [20]. Although the achieved accuracy does not improve the already strong constraints on the real part of the couplings (left panels), the result is consistent with the existing data and increases confidence in the validity of the extraction of $R$. For the imaginary part (right panels), the new experimental value of the $R$ coefficient significantly constrains scalar couplings beyond the limits from all previous measurements. The result is consistent with the SM.

## 15.3 Outlook – the BRAND project

The successful determination of two transverse components of the polarization of electrons emitted in neutron decay in a pioneering and nearly optimal experiment led to the following conclusions: (i) it seems quite possible to decrease the systematic uncertainty by an order of magnitude using existing techniques, (ii) the transverse electron polarization can be studied in a more systematic way by correlating it with the electron momentum, the neutron spin, and also with the recoil proton momentum by constructing larger and higher acceptance detecting systems like e.g. proposed by [21] and operating with the highest intensity polarized cold neutron beam available. In this way, one can study seven correlation coefficients: $H, L, N, R,$

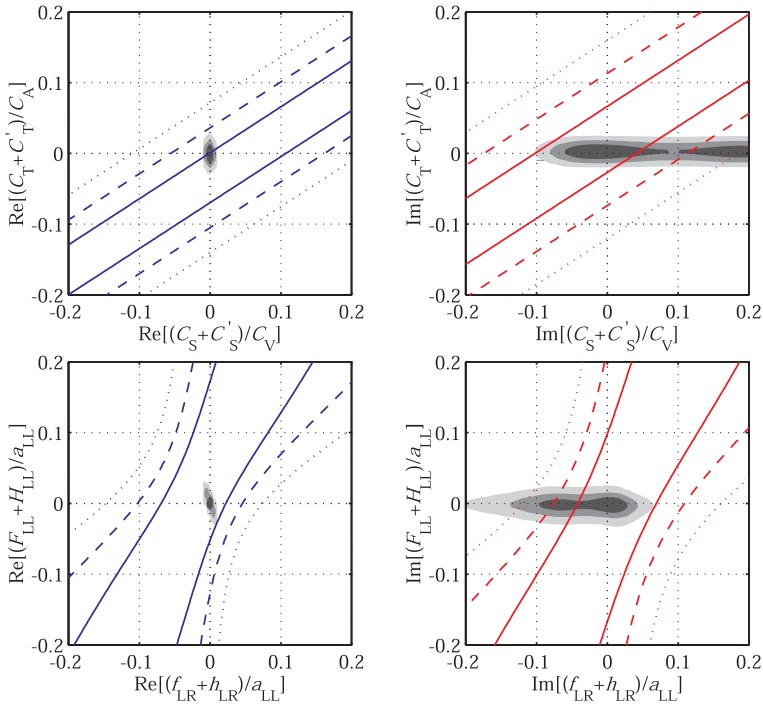

Figure 15.4: Experimental bounds on the scalar vs. tensor normalized couplings (upper) and leptoquark exchange helicity projection amplitudes (lower panels) published in [12]. The gray areas represent the available to date empirical information as listed in [19], while the lines represent the limits resulting from the present experiment. Solid, dashed and dotted lines correspond to 1-, 2- and 3- sigma confidence levels, respectively, in analogy to decreasing intensity of the grey areas.

$S$, $U$ and $V$ where five of them ($H$, $L$, $S$, $U$, $V$) have never been experimentally studied:

$$
\begin{aligned}
\omega(E_e, \Omega_e, \Omega_{\bar{\nu}}) \propto{} & 1 + \\
& a\,\frac{\mathbf{p}_e \cdot \mathbf{p}_{\bar{\nu}}}{E_e E_{\bar{\nu}}} + b\,\frac{m_e}{E_e} + \frac{\langle \mathbf{J} \rangle}{J} \cdot \left[ A\,\frac{\mathbf{p}_e}{E_e} + B\,\frac{\mathbf{p}_{\bar{\nu}}}{E_{\bar{\nu}}} + D\,\frac{\mathbf{p}_e \times \mathbf{p}_{\bar{\nu}}}{E_e E_{\bar{\nu}}} \right] + \\
& \boldsymbol{\sigma}_{\perp} \cdot \left[ H\,\frac{\mathbf{p}_{\bar{\nu}}}{E_{\bar{\nu}}} + L\,\frac{\mathbf{p}_e \times \mathbf{p}_{\bar{\nu}}}{E_e E_{\bar{\nu}}} + N\,\frac{\langle \mathbf{J} \rangle}{J} + R\,\frac{\langle \mathbf{J} \rangle \times \mathbf{p}_e}{J E_e} + \right. \\
& \left. \quad S\,\frac{\langle \mathbf{J} \rangle}{J} \frac{\mathbf{p}_e \cdot \mathbf{p}_{\bar{\nu}}}{E_e E_{\bar{\nu}}} + U\,\mathbf{p}_{\bar{\nu}} \frac{\langle \mathbf{J} \rangle \cdot \mathbf{p}_e}{J E_e E_{\bar{\nu}}} + V\,\frac{\mathbf{p}_{\bar{\nu}} \times \langle \mathbf{J} \rangle}{J E_{\bar{\nu}}} \right],
\end{aligned}
\tag{15.10}
$$

where $\boldsymbol{\sigma}_{\perp}$ represents a unit vector perpendicular to the electron momentum $\mathbf{p}_e$ and $J = |\mathbf{J}|$. $\mathbf{p}_{\bar{\nu}}$ and $E_{\bar{\nu}}$ are the antineutrino momentum and energy, respectively.

The coefficients relating the transverse electron polarization to $\mathbf{p}_e$, $\mathbf{p}_{\bar{\nu}}$ and $\mathbf{J}$ have several interesting features. They vanish for the SM weak interaction, and reveal the variable size of the electromagnetic contributions. For $H$ and $N$, the electromagnetic contributions are of the order of 0.06, which can be used for an internal sensitivity check of the Mott polarimeter. Finally, the dependence on the real and imaginary parts of the scalar and tensor couplings alternates exclusively from one correlation coefficient to another with varying sensitivity. This feature allows a complete set of constraints to be determined from the neutron decay alone.

The idea of implementing such a complex measurement was proposed in [22]. An updated version of the measurement can be found in [23]. Presently, the first test run devoted to the verification of the applied detectors and techniques has been completed on the PF1B cold neutron beam at the Laue Langevin Institute in Grenoble, France (ILL).

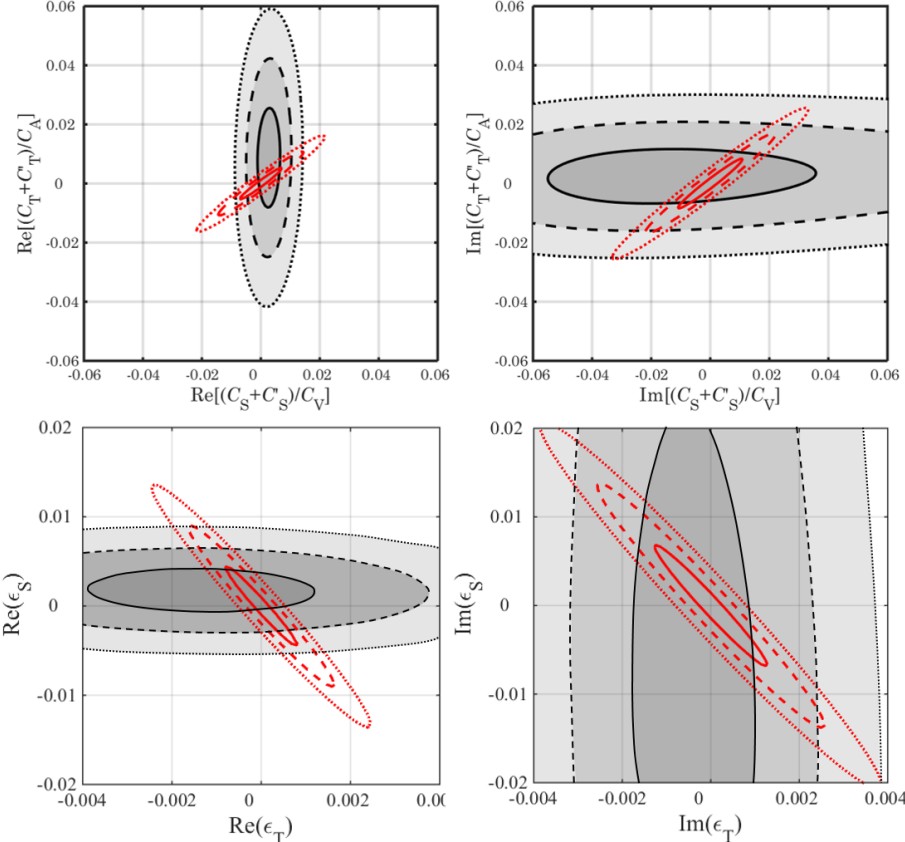

**Figure 15.5:** Experimental bounds on the scalar vs. tensor couplings $\mathfrak{S}$, $\mathfrak{T}$ from (15.2) (upper panels) and translated to EFT parameters $\epsilon_S$, $\epsilon_T$ (lower panels) published in [23]. The gray areas represent the information deduced from available experiments as listed in [24], while the red lines represent the limits resulting from the correlation coefficients $H$, $L$, $N$, $R$, $S$, $U$ and $V$ measured with the anticipated accuracy of $5 \times 10^{-4}$. Solid, dashed and dotted lines correspond to 1-, 2- and 3-$\sigma$ confidence levels, respectively, in analogy to decreasing intensity of the grey areas.

## 15.4 EFT parameterization

In order to permit for sensitivity comparison of low-energy charged-current observables with measurements carried out at high-energy colliders, the model-independent effective field theory (EFT) framework is employed. This approach bridges the classical $\beta$-decay formalism with high-energy physics. The effective nucleon-level couplings $C_i$, $C_i'$ ($i \in [V, A, S, T]$) can be generally expressed as combinations of the quark-level parameters $\epsilon_i$, $\tilde{\epsilon}_i$ ($i \in [L, R, S, T]$) [25]. The real parts of the scalar and tensor couplings parameterize CP-conserving and imaginary parts – CP-violating contributions. The high energy BSM physics quantity that can be compared with $\beta$-decay observables is the cross section for electrons and missing transverse energy (MET) in $pp \to e\bar{\nu} + MET + \ldots$ channel. Both have the same underlying partonic process: $\bar{u}d \to e\bar{\nu}$. Anticipating the experimental accuracy of about $5 \times 10^{-4}$ for the transverse electron polarization related correlation coefficients in the BRAND experiment one would obtain significantly tighter bounds on the real and imaginary parts of scalar and tensor coupling constants and, consequently, on $\epsilon_S$ and $\epsilon_T$ as shown in Figure 15.5. It should be noted that such limits would be competitive to those extracted from the analysis of 20 fb$^{-1}$ CMS collaboration data collected at 8 TeV [26, 27] and even to the planned measurements at 14 TeV.

## Acknowledgments

This work has been supported in part by The National Science Centre, Poland, under the grant No. 2018/29/B/ST2/02505.

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
