# Peer review of "Measurement of the transverse polarization of electrons emitted in neutron decay – nTRV experiment"

_SciPost Physics Proceedings, doi:SciPost Phys. Proc. 5, 015 (2021)_

## Round 1 · Referee Report · Anonymous (Referee 1) · 2021-4-21

Strengths

The paper reviews an experiment measuring for the first time two beta decay correlation coefficients , N and R, connected to the transverse polarisation of the emitted electron. It is clearly written and nicely combines the past experiment and future prospects of a similar experiment BRAND.

Weaknesses

The review is a combination of three existing articles and provides no new insights, apart from updating the literature where appropriate.

Report

The review is part of a series of reviews reporting on particle physics experiments which have been performed at PSI. For this series it is very favorable also including this unique experiment, however, the present manuscript can not be excepted due to self-plagiarism. Only a few passage are new, the dominant part of the review is a combination of three articles:
Kozela et al., PRL102,172301(2009)
Kozela et al., PRC85,045501 (2012)
Bodek et al., EPJ Web Conf 219, 04001 (2019)

For a publication the majority of the paper has to be reformulated, otherwise copyright issues might arise.
This is also true for all images, which currently are not correctly referenced.

As the paper is a combination of previously reviewed original research paper, no revision is required.
However, some passaged highlighted in yellow would profit from an additional sentence for clarification.

Requested changes

Reformulation of currently unaltered extract of previous papers.
The passages which are copied are highlighted in different colors as function of there origin, in the attached document.

Attachment

---

## Round 2 · Author Response

The article is meant to recall the essential results of the pioneering experiment nTRV as a base for a new project. This is why several description parts taken from original publications were only slightly modified. Figure reproduction rights were checked. The abstract has been rewritten for the most and all the Reviewer's suggestions were addressed both in the text, in figures and in the Bibliography. One more citation was added. One major question concerning the Mott vertex position distribution was answered, too.

---

## Round 2 · List of Changes

1. Abstract got a new formulation.
  2. The structure of the article remains unchanged.
  3. Appropriate citations have been added in all reproduced Figures.
  4. In Figure 1, the Mott scattering vertex was drawn in red.
  5. A new citation has been added in order to clarify Figure 15.5.

---

## Round 3 · List of Changes

The questioned parts of the text were rephrased in large extent.

You are currently on this page

Resubmission scipost_202103_00021v3 on 8 July 2021
Resubmission scipost_202103_00021v2 on 8 June 2021

---

## Editorial Decision

published